# COVID-19 Vaccination and Mental Health Outcomes among Greek Adults in 2021: Preliminary Evidence

**DOI:** 10.3390/vaccines10081371

**Published:** 2022-08-22

**Authors:** Kyriakos Souliotis, Lily E. Peppou, Theodoros V. Giannouchos, Myrto Samara, Dimitra Sifaki-Pistolla, Marina Economou, Helena C. Maltezou

**Affiliations:** 1Faculty of Social and Political Sciences, University of Peloponnese, 221 00 Corinth, Greece; 2Health Policy Institute, 151 23 Athens, Greece; 3Unit of Social Psychiatry & Psychosocial Care, University Mental Health, Neurosciences & Precision Medicine Research Institute, “Costas Stefanis” (UMHRI), 156 01 Athens, Greece; 4Prefrecture of Athens Mental Health Promotion Programme, First Department of Psychiatry, Medical School, Aiginition Hospital, National & Kapodistrian University of Athens, 157 72 Athens, Greece; 5Department of Health Services Policy & Management, Arnold School of Public Health, University of South Carolina, Columbia, SC 29208, USA; 6Third Department of Psychiatry, Faculty of Medicine, School of Health Sciences, Aristotle University of Thessaloniki, 541 24 Thessaloniki, Greece; 7Directorate for Research, Studies and Documentation, National Public Health Organization, 151 23 Athens, Greece

**Keywords:** vaccination, COVID-19, mental health

## Abstract

Existing research on the association between COVID-19 vaccination and quantitatively measured mental health outcomes is scarce. We conducted a cross-sectional telephone survey on a random sample of 1039 adult Greek citizens in June 2021. Among the participants, 39.6% were vaccinated with two doses, 23.1% with one dose, 21.4% were planning to become vaccinated later, and 8.1% refused vaccination. Compared to those fully vaccinated, those against vaccination (“deniers”) and those who planned to do so later on (“not vaccinated yet”) had significantly higher scores across three stress, anxiety, and depression construct scales. Our findings suggest an association between COVID-19 vaccination status and mental health.

## 1. Introduction

The ongoing coronavirus disease 2019 (COVID-19) pandemic has impinged on the mental health of the population worldwide, although the evidence is still emerging. Moreover, longitudinal data suggest a reciprocal association between mental disorders and COVID-19 [1]. Consequently, the World Health Organization (WHO) has acknowledged that addressing existing and emerging mental health needs is an integral part of effectively responding to the pandemic [2].

Mental health seems to be linked to the pandemic due to the major changes that were required in people’s daily lifestyle [3]. In most countries, complete or partial lockdowns were implemented, with restrictions targeting mobility, prohibition of non-essential activities, working from home, vaccination against COVID-19, and others. These precautionary measures were enforced for the first time in modern society in most countries, worldwide. Consequently, they had a severe impact on daily life, social activities, wellbeing, and health [4,5,6].

Psychological variables, such as anxiety and distress levels have been found to be associated with people’s adherence to precautionary public health measures [2,7,8]. Nonetheless, some studies have indicated that elevated anxiety is linked to better compliance with public health measures [7,9], while others link it with poorer compliance [8]_._ A possible explanation for this discrepancy pertains to methodological differences across studies, different conceptualizations of mental health issues, as well as observed heterogeneity in precautionary measures [10].

Vaccination against COVID-19 is a major public health measure, especially for curbing the spread of the virus [11]. Despite this, a recent review has reported noteworthy rates of vaccine hesitancy, with factors such as socio-demographic characteristics and individuals’ perceptions shaping refusal rates [12,13]. Overall, knowledge, attitudes, and practices towards the disease itself have been associated with coherence to the recommended precautionary measures, including vaccinations [14,15,16]. However, the association between mental health and vaccination, measured as actual, rather than intended, behavior has not been fully explored; with some evidence, nonetheless, indicating that positive attitudes to vaccination and mental health are strongly interrelated [7,9].

In this context, the aim of the present study is to explore the association between COVID-19 vaccination status and mental health outcomes (i.e., stress, anxiety, and depression) during the third wave of the pandemic on a random and representative sample of Greek adults. It merits noting that this report is a part of a larger study investigating the mental health impact of the ongoing pandemic on the Greek population.

## 2. Methods

A random sample of 1205 adult Greek citizens participated in a cross-sectional telephone survey during the time period 31 May 2021–13 June 2021. A sample size estimation had been performed, by hypothesizing a confidence level of 95%, 5% margin of error for an overall population size of 8,693,742 (Greek citizens over 18 years old based on the 2011 National Population Census). The estimated sample size was 385; however, we decided to increase it (by more than three-fold) and therefore targeted to 1205 participants with a pre-defined accepted response rate of 80%. Of the 1205 adults, 1039 were finally recruited (response rate = 86.2%).

Inclusion criteria were fluency in Greek language and being 17 years of age or older. The sample was distributed proportionately among the 13 administrative regions in the country in accordance with the 2011 National Population Census. Potential participants were identified using a random stratified selection process from the 2011 national landline telephone directory. Within each household, the persons who had their birthday most recently were selected to answer and at least 6 callbacks were made when an individual did not answer. Participants gave consent verbatim. The study received approval from the Ethics Committee of the Health Policy Institute and was performed in accordance with the ethical standards delineated in the Declaration of Helsinki 1964/2013.

Information was garnered in the form of a fully structured telephone interview that consisted of various sections; however, some of them are beyond the scope of the current report and are not presented.

With respect to the current report, the study outcome was participants’ current mental health status across three dimensions: stress, anxiety, and depression. These were measured using the self-reported Depression, Anxiety and Stress Scale (DASS-21), which has been validated in the Greek population and used in previous work [17,18]. The DASS-21 entails three factors, each of which ranges from 0 to 42 points. Based on cut-off scores, there are five different severity labels for each subscale (normal, mild, moderate, severe, and extremely severe) (Table 1).

The main independent variable was participants’ current vaccination status. Participants were asked “Have you been vaccinated for COVID-19?” with the mutually exclusive options being: “Yes, with both doses” (i.e., full vaccination), “Yes, with one dose” (i.e., partial vaccination, as at the time single-shot vaccines were not yet available in the country), “Not yet, but will get vaccinated later” (respondents who were not yet vaccinated for practical reasons, vaccine availability issues, or fear of the vaccine), “No, I am a vaccine denier”, or “Other/None of the above”. We also obtained information on sociodemographic (age, gender, area of residence marital status, educational level, income, occupation), physical characteristics (self-reported health status, body mass index), and family (household size, underage children living in household, household member vulnerable to COVID-19), and perceptions of SARS-CoV-2 and COVID-19 pandemic (specifically “COVID-19 is here to stay”, “the coronavirus is unpredictable”, and “the coronavirus is fabricated”), to control for associations between these variables and the outcomes of interest, based on findings from previous work [19,20].

Prior to beginning data collection, the interview schedule was pilot-tested for its comprehensibility on a convenience sample of 50 adults from various age groups and with different educational backgrounds. Some amendments were made to the instrument to facilitate its intelligibility. Furthermore, before proceeding to the analysis, we checked overall reliability of the tool (i.e., including the DASS-21 questions and the additional questions on vaccination status, the socio-demographic, physical and family characteristics of the sample, as well as their perceptions about COVID-19 and the virus). We obtained satisfying rates, with Cronbach’s alpha = 0.91.

Statistical significance was evaluated using Pearson’s χ^2^ and Kruskal–Wallis tests. Two-tailed tests were used, and statistical significance was considered at a *p*-value  <  0.05. Three multivariable generalized linear regression models were used to estimate the association between respondents’ vaccination status and the DASS-21 constructs separately, when controlling for covariates described above. Geographic-level fixed effects and clustered standard errors at the geographical region of residence were used to control for unobserved time-invariant regional characteristics. Additional sensitivity analyses were conducted to address missing data in control variables and to evaluate the robustness of our findings. Data were collected in SPSS and statistically analyzed in Stata (version 17.0; StataCorp, College Station, TX, USA).

## 3. Results

Among the 1039 participants, 719 fully completed the questionnaire and were included into the analysis. Overall, 39.6% were reported to have been vaccinated with both COVID-19 vaccine doses, and 23.1% with one dose. About one-fifth of participants were planning to be vaccinated later, while 8.1% were against COVID-19 vaccination. The overall mean DASS-21 Stress, Anxiety, and Depression subscores were 12.3 (standard deviation (st.d.) = 9.7), 6.5 (st.d. = 8.2), and 10.0 (st.d. = 10.3), respectively. Severe or extremely severe scores were reported by around 13% of participants in the Stress and Anxiety scales and 17% in the Depression scale. In the stratified analysis, participants who had been vaccinated with both doses of the COVID-19 vaccine had significantly lower scores across all three mental health subscales compared to individuals with all other vaccination statuses (Figure 1).

The multivariate adjusted regression analyses revealed that, compared to those fully vaccinated, deniers of vaccination had significantly higher scores (6.37 (95% confidence interval = 3.69–9.04, *p*-value < 0.001), 2.55 (95% confidence interval = 0.26–4.84, *p*-value = 0.029), and 5.84 (95% confidence interval = 2.90–8.77, *p*-value <0.001)) across the Stress, Anxiety, and Depression subscales respectively. Similar associations were also observed across all three mental health subscales for those who were not yet vaccinated but were planning to be vaccinated later compared to vaccinated individuals. We also observed an increased score by 2.6 points on the Depression subscale for individuals who had received only one dose of the vaccines, compared to those who were vaccinated with both doses (adjusted coefficient = 2.63, 95% confidence interval = 1.53–3.73, *p*-value < 0.001). The findings were similar in supplemental sensitivity analyses handling missing data.

## 4. Discussion

Our results indicate an association between COVID-19 vaccination and selected mental health outcomes. Fully vaccinated participants displayed the lowest levels of stress, anxiety, and depression, in line with evidence suggesting that adherence to precautionary measures alleviates mental health symptoms [8,21]. Low levels were also observed in individuals in the “none of the above” category, which consists mostly of persons previously infected by SARS-CoV-2. Interestingly, even a single dose of COVID-19 vaccine was associated with lower stress and anxiety levels compared to no vaccination. Therefore, people with evidence of immunity (either through natural infection or vaccination) seem to have better mental health outcomes, as compared to unvaccinated persons. Stress and anxiety can be conceptualized as reactions to threat (real or perceived, respectively) [22]. Therefore, as COVID-19 vaccine offers major protection from life-threatening illness, it may reduce the level of actual and perceived threat, thus ameliorating stress and anxiety symptoms.

Interestingly, while for stress and anxiety, one dose of vaccination was enough for alleviating these symptoms, this was not the case for depression. Partially vaccinated people scored high in depression as compared to those fully vaccinated. A potential explanation is the restrictions still in place for those partially vaccinated and the accompanying loss of freedom and normality. In the biopsychosocial model of depression, loss in its various forms has a central position in explaining the syndrome [23]. Thus, fully vaccinated individuals were potentially in a better position to reclaim their lives and return to normality.

Contrarily, unvaccinated people were still facing the unprecedented disruption of daily activities in the context of COVID-19 pandemic uncertainties and fear of infection. Importantly, the group “not vaccinated yet” entailed people who could had not been able to receive the vaccination, rendering them vulnerable to the psychosocial consequences of the pandemic without being able to do resolve this issue. Moreover, evidence indicates that existing mental health symptoms may be barriers to vaccination [24,25,26]; thus, the category “not vaccinated yet” might have also consisted of individuals who already experienced fear and uncertainty about vaccines and therefore their stress, anxiety, and depression levels were already heightened. Finally, deniers were also susceptible to poorer mental health, possibly due to the social marginalization they might experience as anti-vaxxers.

Additionally, as mentioned before, knowledge and overall attitude towards the disease have been found to influence compliance to COVID-19-related measures, including vaccinations [4,16]. For example, in the study of Galle and colleagues, the authors stressed that satisfactory levels of knowledge were associated with compliance to the measures [16]. At the same time, knowledge is also linked to better mental health outcomes [27], indicating a potential confounding effect of knowledge in the present design. We tried to address this by controlling for perceptions about the virus, which are interlinked to knowledge [27]; however, future study should entail a health literacy assessment.

The contribution of knowledge and perceptions about the virus in shaping health behaviors stress the importance of information and awareness-raising campaigns prior to implementation of any measures. The value of vaccinations and their association with mental health should be in embodied to national information strategies against COVID-19. The effectiveness of preventive measures and warnings and messages issued by public health authorities should be easy to understand, concise, and convincing [28]. Concomitantly, evidence from this study indicates that vaccination can also lead to better mental health outcomes, preventing in this way the spread of a silent mental health pandemic [29]; however, better mental health also encourages adherence to precautionary measures. In this way, prioritizing the mental health of the population is also a way to tackle the adverse psychosocial effects of the ongoing pandemic. Integrating mental health in primary health care, digitalization of services, the emphasis on mental health promotion activities and on building community resilience are among the strategies that have been recommended to this end [19].

The study had several limitations that should be discussed. The reported data on fully vaccinated (39.6%) and one-dose vaccinated individuals (23.1%) may have been over or underestimated comparing to the national records on vaccination metrics (30% and 40%, respectively). This can be attributed to various reasons, including different time frames of data collection (our survey was conducted during the first two weeks of June), information bias due to self-report (e.g., social desirability bias) or sampling bias of the study. The latter perhaps explains the discrepancy in fully vaccinated rates, as people of lower socio-economic or educational background, people with immigrant status, or people with severe mental health issues might have been inaccessible by phone or refused to participate, resulting in non-participation bias. As this study is a part of larger research effort to monitor, measure, and investigate in depth the mental health impact of the COVID-19 pandemic in the Greek population, feasibility issues prevented us from adopting a more multifaceted and thorough sampling strategy.

Moreover, the cross-sectional design could not elucidate the direction of the association between mental health and vaccination and thus causal inferences cannot be made. A cohort study is warranted so as to shed light on the direction of causality. Concomitantly, the study did not control for a pre-existing mental health condition due to difficulties in obtaining valid responses. Mental disorders are particularly stigmatized in Greece [30] and, as a result of this, people might be reluctant to disclose this. Despite its limitations, this is the first study to have explored the association between COVID-19 vaccination status and mental health in a random and representative sample of the general population and to suggest an association between COVID-19 vaccination status and mental health. The current findings and identified trends can mainly serve policy makers in Greece by informing vaccination policies and effective prevention strategies.

## Figures and Tables

**Figure 1 vaccines-10-01371-f001:**
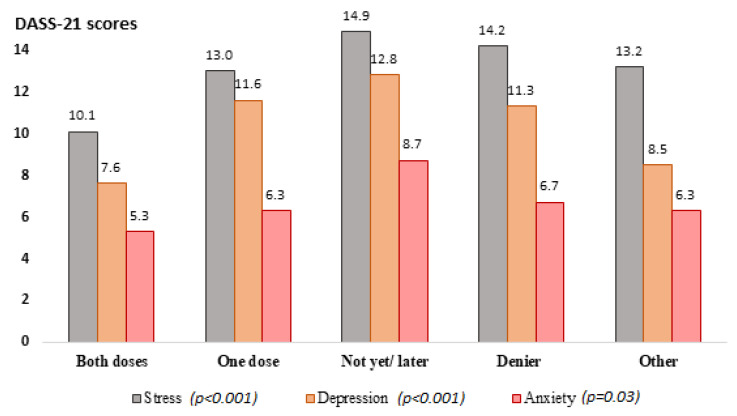
DASS-21 scores for the Depression, Anxiety, and Stress constructs by participants’ vaccination status. Colors should be used for this figure in print.

**Table 1 vaccines-10-01371-t001:** Cut-off scores for DASS-21.

	Depression	Anxiety	Stress
Normal	0–4	0–3	0–7
Mild	5–6	4–5	8–9
Moderate	7–10	6–7	10–12
Severe	11–13	8–9	13–16
Extremely severe	14+	10+	17+

## Data Availability

Data could be available upon request from the corresponding author, only for research purposes. Access to the primary data could not be feasible due to data privacy reasons.

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
