# Peer review of "COVID-19 Vaccination and Mental Health Outcomes among Greek Adults in 2021: Preliminary Evidence"

_vaccines, 2022, doi:10.3390/vaccines10081371_

Round 1

Reviewer 1 Report

This is a brief, well-written report with a reasonably good subject number. The results, however, now seem very generalized. The consequences of COVID-19 vaccination pre- and post-vaccination are well documented elsewhere. 

The occurrence of some of these adverse events is consistent with what is already known about the vaccines from clinical trials. It may take longer to see symptoms like neurological symptoms or mental health conditions post covid -19 vaccination. 

The authors should revise figure 1; the text is confusing. 

Author Response

We would like to thank the reviewer for his/her comments. Indeed, this is a short report with general trends on the topic, since it's the first study with Greek data. In one of our next studies, we will include more analytical and specific outcomes. However, we have revised the discussion according to other reviewers' comments and discussed all these issues. 

As regards, the comment about figure 1, we have now revised it and made it clearer. Lastly, we have made a linguistic check and revisions to make the entire text even clearer and provide potential explanations etc. Thank you for reviewing once again. 

Reviewer 2 Report

First of all I would like to thank for the opportunity to review this paper. COVID-19 is an ongoing pandemic that has resulted in global health, economic and social crises. This had great impact on people’s behaviors, relationship and overall mental health. Less studied is the association between COVID-19 vaccination and mental health outcomes. In this context, aim of the paper under review is to evaluate the association between COVID-19 vaccination status and mental health outcomes in a sample of the general population in Greece.

The subject under study is certainly important, especially in the historical period we are experiencing. The article presents interesting results but, but it need a major revision before publication, especially for its local impact and some methodological concerns.

Title: it can be improved, highlight the object of the study: place, time and person.

Introduction: It must be improved, the authors should make clearer what is the gap in the literature that is filled with this study. The authors must better frame their study within the vast body of literature that addressed the issue of COVID-19 related knowledge and vaccination (refer to articles with DOI: https://doi.org/10.3390/ijerph182010872) also at international level, since knowledge can be a bias factor in the evaluation between vaccination and mental health.

Methods: It is not clear how the survey was conducted: was it a telephone interview of a self-filled questionnaire? in the last case, how did the sample get the questionnaire? The used questionnaire was in part a standardized one, but in part the Authors used non-standard questions. A validation process must be performed to evaluate the modified tool. What about reliability, intelligibility and validation index?

When the study was performed? The period of the study seems missing.

The number of enrolled people is large but the authors do not propose a minimum sample size, what is the reference population? How large is it? Without the numerical identification of the reference population is not clear the validity of the study. A non-representative sample is by its self a non-sense-survey.

Statistical analysis: I suggest to insert a measure of the magnitude of the effect for the comparisons. Please consider to include effect sizes.

Figure 1 must be improved.

Discussion: I also suggest expanding. Emphasize the contribution of the study to the literature. The discussion must be updated with the discussion on knowledge as possible confounding factor, a paragraph should be added with a proper reference (see the above mentioned reference). The Authors should add more practical recommendations for the reader, based on their findings. Also, the section of limitations and future search is also very short, the Authors could elaborate on that.

Author Response

First of all I would like to thank for the opportunity to review this paper. COVID-19 is an ongoing pandemic that has resulted in global health, economic and social crises. This had great impact on people’s behaviors, relationship and overall mental health. Less studied is the association between COVID-19 vaccination and mental health outcomes. In this context, aim of the paper under review is to evaluate the association between COVID-19 vaccination status and mental health outcomes in a sample of the general population in Greece.The subject under study is certainly important, especially in the historical period we are experiencing. The article presents interesting results but, but it need a major revision before publication, especially for its local impact and some methodological concerns.

Response: We would like to thank you for all remarks and comments. We tried to manage all of them and revise the manuscript accordingly.

Title: it can be improved, highlight the object of the study: place, time and person.

Response: We revised the title as follows “COVID-19 vaccination and mental health outcomes among Greek adults in 2021: preliminary evidence.”

Introduction: It must be improved, the authors should make clearer what is the gap in the literature that is filled with this study. The authors must better frame their study within the vast body of literature that addressed the issue of COVID-19 related knowledge and vaccination (refer to articles with DOI: https://doi.org/10.3390/ijerph182010872) also at international level, since knowledge can be a bias factor in the evaluation between vaccination and mental health.

Response: We would like to thank the reviewer for his/her valuable input. We have tried to address his/her point, by adding literature in the introduction, given the word limit of short reports. Nonetheless, we think he/she might have missed a piece of information in Methods. In our study we did not measure knowledge per se (we had done so in our previous study, see Archives of Hellenic Medicine, 39(3), May-June 2023, 354-365, https://www.mednet.gr/archives/2022-3/354abs.html due to ceiling effect of those items in the previous survey, especially the ones about modes of transmission). Nonetheless, we did entail questions measuring participants’ level of agreement/disagreement with certain beliefs about the virus and the pandemic (e.g. “it is here to stay”, “the virus is unpredictable”, “the virus is fabricated” etc.). Therefore, we have included a proxy of knowledge (as perceptions and beliefs about the virus are influenced by knowledge among others) and we have controlled for these variables in the regression analysis (see Methods, “We also obtained information on sociodemographic (age, gender, area of residence marital status, educational level, income, occupation), physical (self-reported health status, Body Mass Index), and family (household size, underage children living in household, household member vulnerable to COVID-19) characteristics, and perceptions of SARS-CoV-2 and COVID-19 pandemic (specifically “COVID-19 is here to stay”, “the coronavirus is unpredictable” and “the coronavirus is fabricated”), to control for associations between these variables and the outcomes of interest, based on findings from previous work [22,23]”. However, we thank the reviewer for placing emphasis on this and we have included the importance of knowledge in adherence to precautionary measures and vaccination in the introduction.

Moreover, we have added a paragraph linking the pandemic to mental health through the implementation of measures and other changes in lifestyle. (Page 1, Lines 37-42 and page 2, lines 54-56)

Methods: It is not clear how the survey was conducted: was it a telephone interview of a self-filled questionnaire? in the last case, how did the sample get the questionnaire? The used questionnaire was in part a standardized one, but in part the Authors used non-standard questions. A validation process must be performed to evaluate the modified tool. What about reliability, intelligibility and validation index?

Response: We would like to thank the reviewer for the points he/she raises. First of all, we have clarified in the last paragraph of the introduction that the present short report is a part of a larger study whose aim was to investigate the mental health impact of the COVID-19 pandemic and its determinants on a random and representative sample of Greek people.  Therefore, the interview schedule consisted of many sections, which in turn entailed various scales. All the scales were well-validated tools, especially the one that taps mental heath outcomes. We have not included the other scales (e.g. Philadelphia Mindfulness scale, Self-Compassion Scale, Warwick- Edinburgh Mental Well-being scale) in the description of the methods, in order to avoid readers’ confusion (as germane results are not presented in the Results section). We hope that the reviewer will see our point about this. Nonetheless, in the third paragraph of the Methods, we have tried to elaborate on the points he/she is raising.

The DASS-21 scale, which measures the main study outcome is a widely used instrument in Greece with established psychometric properties. Moreover, prior to data collection, the interview schedule was pilot tested for comprehensibility on a convenience sample (snowballing technique) of 50 members of the general population. We tried to test it on people belonging to different age groups and with various educational backgrounds. Some amendments took place (especially rephrasing certain questions to make them more intelligible) prior to the beginning of the survey. Moreover, overall reliability of the tool was tested by us, although the DASS-21 was already a validated tool.

 All this information has been added in the Methods section.

When the study was performed? The period of the study seems missing.

Response: We have already mentioned that in lines 65-66 . “A random sample of 1205 adult Greek citizens participated in a cross-sectional tele-phone survey during June 2021.” However, we have clarified it further (exact dates) to be sure that we are meeting the reviewer’s point.

The number of enrolled people is large but the authors do not propose a minimum sample size, what is the reference population? How large is it? Without the numerical identification of the reference population is not clear the validity of the study. A non-representative sample is by its self a non-sense-survey.

Response: Thank you for this remark. We had indeed performed a sample size estimate but due to the brief report shorter lengths we hadn’t include it in the test. We have now added it and revised the methods accordingly. We specifically mention that: “A sample size estimation had been performed, by hypothesizing a confidence level of 95%, 5% margin of error for an overall population  size of 7,122.830 (Greece citizens over 18 years old based on the 2011 National Population Census). The estimated sample size was 385 and we decided to increase it (triple it or more) and therefore, targeted to 1205 participants with a pre-defined accepted response rate of 80%. Of the 1205 adults, 1039 were finally recruited (response rate = 86.2%. (Lines 66-71).

Please note that sample size calculation has been performed in line (and in harmony) with previous telephone (mental) health surveys in the country, which have recruited random and representative samples of the Greek population amid the pandemic (e.g. Souliotis et al., 2021, https://doi.org/10.1177/00469580211022913, Souliotis et al., 2021; Archives of Hellenic Medicine, 39(3), May-June 2023, 354-365, https://www.mednet.gr/archives/2022-3/354abs.html).

Statistical analysis: I suggest to insert a measure of the magnitude of the effect for the comparisons. Please consider to include effect sizes.

Response: Thank you for your comment. We will consider adding these numbers in a more detailed future paper which will be more analytical, since this was a more descriptive and trends reporting paper.

Figure 1 must be improved.

Response: We have improved it.

Discussion: I also suggest expanding. Emphasize the contribution of the study to the literature. The discussion must be updated with the discussion on knowledge as possible confounding factor, a paragraph should be added with a proper reference (see the above mentioned reference). The Authors should add more practical recommendations for the reader, based on their findings. Also, the section of limitations and future search is also very short, the Authors could elaborate on that.

Response:

Thank you for the remark, we have revised the discussion accordingly

Reviewer 3 Report

Estimated Dr. Souliotis and coworkers,

I've read with great interest the present paper dealing with the topic of COVID-19 vaccination and mental health outcomes. Through a random and large sample that was collected during 2021, Authors have stresses that, compared to the fully vaccinated individuals, those against vaccination (i.e. deniers) and those planning to do so later on had significantly higher scores across three Stress, Anxiety, and Depression construct scales.

The study is both interesting and consistent with the aims of vaccines. However, some improvements are required:

1) Figure 1, the core and the tentpole of this study, is confusing in its design, and must be reformulated by including a proper reporting of the axes and captions;

2) section included in row 112-123 could not be fully appreciated because of some issue in the reporting of values. More precisely, Authors write that: "The multivariate adjusted regression analyses revealed that, compared to those fully vaccinated, deniers of vaccination had 6.37 (95% Confidence Intervals=3.69-9.04, p- value<0.001), 2.55 (95% Confidence Intervals=0.26-4.84, p-value=0.029), and 5.84 (95% 114 Confidence Intervals=2.90-8.77, p-value<0.001), significantly higher scores across the Stress, Anxiety, and Depression subscales respectively". Authors could improved these results by preventively reporting the full scales and ranges either in the material and methods or in the results.

Author Response

Estimated Dr. Souliotis and coworkers,

I've read with great interest the present paper dealing with the topic of COVID-19 vaccination and mental health outcomes. Through a random and large sample that was collected during 2021, Authors have stresses that, compared to the fully vaccinated individuals, those against vaccination (i.e. deniers) and those planning to do so later on had significantly higher scores across three Stress, Anxiety, and Depression construct scales.

The study is both interesting and consistent with the aims of vaccines. However, some improvements are required:

Response: We would like to thank you for all remarks and comments. We tried to manage all of them and revise the manuscript accordingly.

1) Figure 1, the core and the tentpole of this study, is confusing in its design, and must be reformulated by including a proper reporting of the axes and captions;

Response: We’ve enhanced it.

2) section included in row 112-123 could not be fully appreciated because of some issue in the reporting of values. More precisely, Authors write that: "The multivariate adjusted regression analyses revealed that, compared to those fully vaccinated, deniers of vaccination had 6.37 (95% Confidence Intervals=3.69-9.04, p- value<0.001), 2.55 (95% Confidence Intervals=0.26-4.84, p-value=0.029), and 5.84 (95% 114 Confidence Intervals=2.90-8.77, p-value<0.001), significantly higher scores across the Stress, Anxiety, and Depression subscales respectively". Authors could improved these results by preventively reporting the full scales and ranges either in the material and methods or in the results.

Response: Indeed, you are right. We have now added this information in the methods (Table 1, Lines 86-89).

Reviewer 4 Report

In this manuscript titled “COVID-19 vaccination and mental health outcomes: evidence from Greece”, the authors explored the association between COVID-19 vaccination and mental health status via a survey in Greece. The authors found that a full vaccination is strongly associated with a better mental health status. Overall, this is a very interesting and timely study. However, the manuscript is not well written and some data look confusing. Therefore, a major revision is needed to improve the quality of this manuscript.

Major issues:

1. Line 19, “There are no published data on the association between COVID-19 vaccination and mental health outcomes”, this statement is confusing and over-simplifies the situation. Nearly every major published work on COVID-19 vaccine clinical trials included the list of adverse events after vaccination which contained information regarding mental health issues. Please rephrase this sentence to avoid confusion.

2. Line 33, “longitudinal data suggests a bidirectional association between mental disorders and COVID-19”, what is the meaning of “bidirectional association”? Please elaborate.

3. Line 48-50, “This is a marked omission given that evidence indicating that mental health and vaccination acceptance are strongly interrelated”, this statement is very confusing. What is the meaning of “marked omission”? If there have already been evidence of “mental health and vaccination acceptance are strongly interrelated”, why do the authors still think there is “marked omission”?

4. Line 97, “Among the 1039 participants, 719 fully completed the questionnaire”. Among the 1039 recruited or responded participants, why is the fully completion rate so low? Could the authors explain it in the manuscript? What are those not fully completed cases? Did they just abruptly end the survey phone call before finishing the full questionnaire?

5. Line 98-99, “Overall, 39.6% reported to have been vaccinated with both COVID-19 vaccine doses, and 23.1% with one dose”. I did an online search via Google, finding that the fully vaccination rate in Greece in June 2021 was about 30% and about 40% people got at least one dose vaccination. It seems the fully vaccinated percentage and one-dose vaccinated percentage are a little higher in the participants than in the whole population in Greece. Could the authors explain the reason and discuss it in the limitations part of this manuscript?

6. Line 147-152, the authors discussed the data from participants who answered “not yet” and vaccine denier. However, it omits the interesting fact that not-yet participants had a higher score than deniers. Is this difference between these two groups statistically significant? If it is, do the authors have potential explanations for this difference?

Author Response

In this manuscript titled “COVID-19 vaccination and mental health outcomes: evidence from Greece”, the authors explored the association between COVID-19 vaccination and mental health status via a survey in Greece. The authors found that a full vaccination is strongly associated with a better mental health status. Overall, this is a very interesting and timely study. However, the manuscript is not well written and some data look confusing. Therefore, a major revision is needed to improve the quality of this manuscript.

Response: We would like to thank you for all remarks and comments. We tried to manage all of them and revise the manuscript accordingly.

Major issues:

  1. Line 19, “There are no published data on the association between COVID-19 vaccination and mental health outcomes”, this statement is confusing and over-simplifies the situation. Nearly every major published work on COVID-19 vaccine clinical trials included the list of adverse events after vaccination which contained information regarding mental health issues. Please rephrase this sentence to avoid confusion.

Response: Thank you for the comment. Our point was that only few studies (at the time we were writing the short report there was actually none) have aimed to primarily investigate the association between vaccination status and ill mental health by employing quantitative measures. So you are quite right, it is not about published or existing data, rather, it is about the design, implementation and write-up of these studies.  Most of the studies just refer to several adverse effects and indirectly assess mental health.

  1. Line 33, “longitudinal data suggests a bidirectional association between mental disorders and COVID-19”, what is the meaning of “bidirectional association”? Please elaborate.

Response: We have used the term bidirectional in line with the longitudinal study that has demonstrated this association: Taquet, M.; Luciano, S.; Geddes, J.R.; Harrison, P.J. Bidirectional associations between COVID-19 and psychiatric disorder: retrospective cohort studies of 62 354 COVID-19 cases in the USA. The Lancet Psychiatry. 2021;8,130–40.

 What it is meant by this, is the following: researchers of this study showed that having COVID-19 at baseline (T0)  was a risk factor for developing a mental disorder at a later stage (T1). At the same time, having a mental disorder at baseline (T0) was a risk factor for having COVID-19 at a later stage (T1) (hence bi-directional, i.e. goes both ways, from COVID-19 to mental illness but also, from mental illness to COVID-19). Nonetheless, we have replaced the term bidirectional, with the word reciprocal. Maybe the reviewer will find the new term more suitable.

  1. Line 48-50, “This is a marked omission given that evidence indicating that mental health and vaccination acceptance are strongly interrelated”, this statement is very confusing. What is the meaning of “marked omission”? If there have already been evidence of “mental health and vaccination acceptance are strongly interrelated”, why do the authors still think there is “marked omission”?

Response: We thank the reviewer for his/her comment. We meant marked omission in the literature and vaccination acceptance was measured through attitudes to COVID-19 vaccination and not actual behavior. We thought it was marked omission because attitudes are not synonymous to actual behaviours, as behaviours are influenced by more variables than simply by attitudes. Nonetheless, we have tried to clarify this bit by elaborating on it and rephrasing it: “However, the association between mental health and vaccination, measured as actual behaviour rather than as intended,  has not been explored; with some evidence, nonetheless,  indicating that positive attitudes to vaccination and mental health are strongly interrelated [7,9].”

  1. Line 97, “Among the 1039 participants, 719 fully completed the questionnaire”. Among the 1039 recruited or responded participants, why is the fully completion rate so low? Could the authors explain it in the manuscript? What are those not fully completed cases? Did they just abruptly end the survey phone call before finishing the full questionnaire?

Response: As we mention in the methods which we revised according you your and another reviewer comment “ A random sample of 1205 adult Greek citizens participated in a cross-sectional telephone survey during June 2021. A sample size estimation had been performed, by hypothesizing a confidence level of 95%, 5% margin of error for an overall population  size of 8,693,742 (Greece citizens over 18 years old based on the 2011 National Population Census). The estimated sample size was 385 and we decided to increase it (triple it or more) and therefore, targeted to 1205 participants with a pre-defined accepted response rate of 80%. Of the 1205 adults, 1039 were finally recruited (response rate = 86.2%. Among the 1039 participants, 719 fully completed the questionnaire and were included into the analysis.” This final number of 719 cases that fully responded to the questionnaire was included in the analysis in order to minimize the missing data in the analysis. It is not expected that this number will affect negatively the results since it is still much higher than the required estimated sample size (n=385). As regards the reasons of the respondents for not fully answering to all the questions, it should be noted that some of them were willing to respond to the survey but not to all the questions, and based on the ethical standards we had to respect their decision.

  1. Line 98-99, “Overall, 39.6% reported to have been vaccinated with both COVID-19 vaccine doses, and 23.1% with one dose”. I did an online search via Google, finding that the fully vaccination rate in Greece in June 2021 was about 30% and about 40% people got at least one dose vaccination. It seems the fully vaccinated percentage and one-dose vaccinated percentage are a little higher in the participants than in the whole population in Greece. Could the authors explain the reason and discuss it in the limitations part of this manuscript?

Response: We would like to thank the reviewer for his/her valuable comment. This discrepancy can be attributed to various reasons, including different time frames for data collection (our survey was conducted during the first two weeks of June), information bias due to self-report (e.g. social desirability bias) or sampling bias of our study. The latter perhaps explains the discrepancy in fully vaccinated rates, as people of lower socio-economic or educational background, people with immigrant status or people with severe mental health issues might have been inaccessible by phone or refused to participate, resulting in non-participation bias. As this study is a part of larger research effort to monitor, measure and investigate in depth the mental health impact of the COVID-19 pandemic in the Greek population, feasibility issues prevented us from adopting a more multifaceted and thorough sampling strategy. In particular, this study was a replication study of a previous one conducted during the first lockdown in Greece (see .Souliotis et al., 2021, https://doi.org/10.1177/00469580211022913, Souliotis et al., 2021; Archives of Hellenic Medicine, 39(3), May-June 2023, 354-365, https://www.mednet.gr/archives/2022-3/354abs.html) and a similar methodology had to be adopted. Moreover, there were also time and money constrains.

  1. Line 147-152, the authors discussed the data from participants who answered “not yet” and vaccine denier. However, it omits the interesting fact that not-yet participants had a higher score than deniers. Is this difference between these two groups statistically significant? If it is, do the authors have potential explanations for this difference?

Response: We had checked for this difference and no statistical significance was found.

Round 2

Reviewer 2 Report

The Authors have revised and improved the manuscript that is now suitable for publication.

Reviewer 4 Report

All concerns resolved.